# Stellate Ganglia and Cardiac Sympathetic Overactivation in Heart Failure

**DOI:** 10.3390/ijms232113311

**Published:** 2022-11-01

**Authors:** Yu-Long Li

**Affiliations:** 1Department of Emergency Medicine, University of Nebraska Medical Center, Omaha, NE 68198, USA; yulongli@unmc.edu; Tel.: +1-402-559-3016; Fax: +1-402-559-9659; 2Department of Cellular & Integrative Physiology, University of Nebraska Medical Center, Omaha, NE 68198, USA

**Keywords:** arrhythmia, autonomic nervous system, cardiac sympathetic activation, heart failure, inflammation, stellate ganglion

## Abstract

Heart failure (HF) is a major public health problem worldwide, especially coronary heart disease (myocardial infarction)-induced HF with reduced ejection fraction (HFrEF), which accounts for over 50% of all HF cases. An estimated 6 million American adults have HF. As a major feature of HF, cardiac sympathetic overactivation triggers arrhythmias and sudden cardiac death, which accounts for nearly 50–60% of mortality in HF patients. Regulation of cardiac sympathetic activation is highly integrated by the regulatory circuitry at multiple levels, including afferent, central, and efferent components of the sympathetic nervous system. Much evidence, from other investigators and us, has confirmed the afferent and central neural mechanisms causing sympathoexcitation in HF. The stellate ganglion is a peripheral sympathetic ganglion formed by the fusion of the 7th cervical and 1st thoracic sympathetic ganglion. As the efferent component of the sympathetic nervous system, cardiac postganglionic sympathetic neurons located in stellate ganglia provide local neural coordination independent of higher brain centers. Structural and functional impairments of cardiac postganglionic sympathetic neurons can be involved in cardiac sympathetic overactivation in HF because normally, many effects of the cardiac sympathetic nervous system on cardiac function are mediated via neurotransmitters (e.g., norepinephrine) released from cardiac postganglionic sympathetic neurons innervating the heart. This review provides an overview of cardiac sympathetic remodeling in stellate ganglia and potential mechanisms and the role of cardiac sympathetic remodeling in cardiac sympathetic overactivation and arrhythmias in HF. Targeting cardiac sympathetic remodeling in stellate ganglia could be a therapeutic strategy against malignant cardiac arrhythmias in HF.

## 1. Introduction

Heart failure (HF) is a major public health problem worldwide, presented by an inability of the heart to provide metabolic demands and perfusion of organs/tissues and characterized by symptoms and signs, including shortness of breath, fatigue, rapid to irregular heart rate, lung crepitations, elevated jugular venous pressure, and peripheral tissue edema [1,2]. HF affects more than 26 million adults worldwide and an estimated 6 million American adults have HF [2,3], in which coronary heart disease (myocardial infarction, MI)-induced HF with reduced ejection fraction (HFrEF) accounts for about 50% of all HF cases [4,5,6,7]. Considering the stable incidence of HF with an annual increase, the actual burden of treatment and diagnosis in patients with HF has obviously exceeded the projected burden in the United States and worldwide, especially accounting for other factors, including an increased comorbidity burden and advancing age of the population [2,8,9]. Despite advances in diagnosis and therapeutic management of HF, HF still has a high morbidity and mortality rate. As a major feature of HF, cardiac sympathetic overactivation [10,11,12,13,14] triggers malignant arrhythmias and sudden cardiac death [15,16,17,18,19,20,21,22,23], which accounts for nearly 50–60% of the mortality in HF patients [20,24,25,26,27,28,29,30,31,32,33,34,35]. The role of cardiac sympathetic hyperactivation in HF is highlighted by the use of β-blockers and cardiac sympathetic denervation as the key approach to the current therapy of HF [36,37,38,39,40,41,42,43]. However, such pharmacological treatment may not be ideal because some studies have demonstrated that β-blockers do not provide satisfactory protection against sudden cardiac death, and some patients are either intolerant or refractory to this therapy [44,45,46,47,48,49,50]. Additionally, despite being an alternative in managing refractory ventricular arrhythmias [38,43,51,52], cardiac sympathetic denervation has adverse complications (including Horner’s syndrome, hyperhidrosis, paresthesia, and sympathetic fight/fight response loss) that severely limit the use of procedures in HF patients [53,54]. These drawbacks have increased the focus on exploring the mechanisms responsible for HF-increased cardiac sympathetic activation and on identifying effective therapeutic interventions, which are crucial for improving prognosis of HF and reducing its mortality.

The regulation of cardiac sympathetic activation is highly integrated by the regulatory circuitry at multiple levels, including afferent, central, and efferent components of the sympathetic nervous system [55,56]. Much evidence, from other investigators and us, has confirmed the afferent and central neural mechanisms causing sympathoexcitation in HF [57,58,59,60,61,62,63,64,65,66,67]. The stellate ganglion is a peripheral sympathetic ganglion formed by the fusion of the 7th cervical and 1st thoracic sympathetic ganglion. As the efferent component of the sympathetic nervous system, cardiac postganglionic sympathetic neurons located in stellate ganglia provide local neural coordination independent of higher brain centers [56,68]. These neurons innervate the heart to regulate cardiac function through neurotransmitters (e.g., norepinephrine, NE) released from cardiac sympathetic nerve terminals [69]. Much evidence from clinical studies and animal experiments has demonstrated that the remodeling of cardiac postganglionic sympathetic neurons in stellate ganglia could contribute to cardiac sympathetic overactivation and malignant ventricular arrhythmias in HF. In this review, therefore, we discuss cardiac sympathetic remodeling in stellate ganglia and potential mechanisms and the role of cardiac sympathetic remodeling in cardiac sympathetic overactivation and arrhythmias in HF.

## 2. Anatomy and Physiology of Stellate Ganglia (Figure 1)

The sympathetic nervous system is one of the two divisions of the autonomic nervous system, the other being the parasympathetic nervous system. The sympathetic nervous system is composed of preganglionic and postganglionic neurons that are involved in signal transmission to regulate a variety of functions in all peripheral organs/tissues. The cardiac preganglionic sympathetic neurons originate in the intermediolateral column of the spinal cord in the thoracic region with the somata located in the gray rami communicantes bilaterally and symmetrically. Their axons are very short and pass through the white rami communicantes to form the synapses with cardiac postganglionic sympathetic neurons located in the lower cervical and upper thoracic paravertebral ganglia, releasing a neurotransmitter, acetylcholine, from cardiac preganglionic nerve terminals [70,71]. Usually, the 7th cervical and 1st thoracic paravertebral sympathetic ganglia fuse into stellate ganglia, and the latter play a key role in a substantial amount of cardiac neurotransmission [71,72,73]. When acetylcholine released from cardiac preganglionic sympathetic nerve endings activates nicotinic acetylcholine receptors on cardiac postganglionic sympathetic neurons in stellate ganglia, the longer cardiac postganglionic sympathetic nerve terminals innervated the heart release some neurotransmitters (such as norepinephrine, neuropeptide Y, and galanin) to regulate the functions of the heart through the activation of adrenergic receptors and other peptide receptors [72,74].

**Figure 1 ijms-23-13311-f001:**
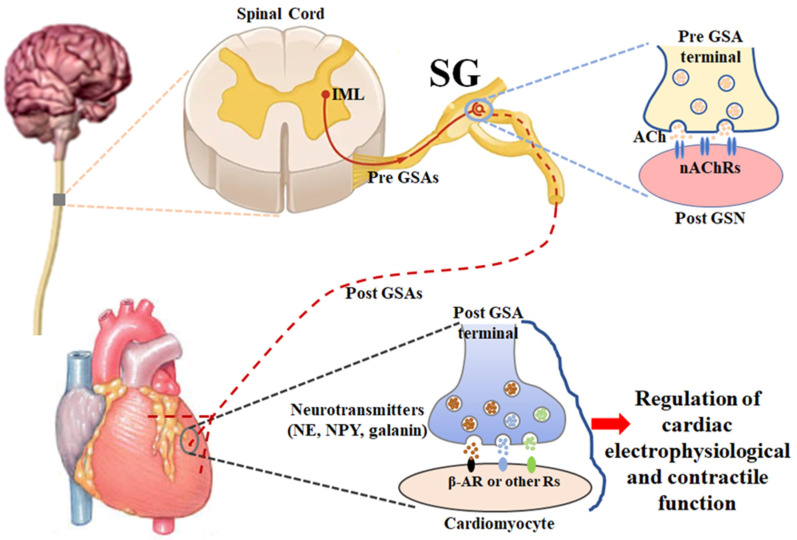
A schematic diagram illustrating the anatomy and physiology of stellate ganglia. ACh: acetylcholine; β-AR: β-adrenergic receptor; IML: intermediolateral nucleus; nAChR: nicotinic acetylcholine receptor; NE: norepinephrine; NPY: neuropeptide Y; Pre GSA: preganglionic sympathetic axon; Post GSA: postganglionic sympathetic axon; Post GSN: postganglionic sympathetic neuron; SG: stellate ganglion.

In the physiological condition, the sympathetic nervous system is responsible for up/downregulating various homeostatic mechanisms in many organs/tissues, especially mediating the fight-or-flight response in situations in which survival is threatened [75,76]. Norepinephrine, which is released from cardiac postganglionic sympathetic neurons in stellate ganglia with their nerve terminals, binds with beta-adrenergic receptors to affect cardiac electrophysiological and contractile functions, including heart rate, cardiac conduction, and myocardial contraction, which finally regulates cardiac output to supply the whole body with oxygenated blood and nutrients [14,75]. It is widely recognized that cardiac postganglionic sympathetic nerve terminals innervate the sinoatrial node, atrioventricular node, His bundle, and contractile myocardium [77]. However, the innervation of the heart with cardiac postganglionic sympathetic neurons in the stellate ganglia presents lateral and regional variations (such as anterior/posterior and left/right divisions of the heart) [78]. In particular, there is an obvious variation and overlap in the innervation of the cardiac tissues from the left and right stellate ganglia [78,79]. The sinoatrial node is primarily innervated by cardiac postganglionic sympathetic nerve terminals from the right stellate ganglion [78]. The conduction system, including the sinoatrial node, atrioventricular node, and His bundle, is more densely innervated than the contractile myocardium [80,81]. Compared to the endocardium of the heart, there is a high density of postganglionic sympathetic innervation on the epicardium of the heart [78,81]. The posterior surface of the heart is mostly innervated by cardiac postganglionic sympathetic nerve terminals from the left stellate ganglion, whereas the anterior surface of the heart is principally innervated by sympathetic nerve terminals from the right stellate ganglion, measured by activation recovery internal (ARI) shortening as a probe of functional innervation [78]. Additionally, it is possible that certain areas of the heart are not innervated by cardiac postganglionic sympathetic nerve terminals from both sides of the stellate ganglia.

Although the amount of neurotransmitters, including norepinephrine, is primarily determined by the intensity of the cardiac postganglionic sympathetic nerve activity, the number of norepinephrine molecules that bind to cardiac adrenergic receptors and induce biological effects on the heart is not only determined by the release of norepinephrine from cardiac postganglionic sympathetic nerve terminals but also by its elimination from the synaptic cleft [82]. Normally, more than 90% of the norepinephrine released into the synaptic cleft is removed by the norepinephrine transporter (NET) [83]. NET (also named the noradrenaline transporter, NAT), a 617 amino acid protein, comprises 12 transmembrane domains at cardiac postganglionic sympathetic nerve terminals [84]. As a member of the sodium/chloride-dependent family of neurotransmitter transporters, NET can take up norepinephrine from the interstitial space to sympathetic nerve terminals with the stochiometric exchange of sodium and chloride against their electrochemical gradient [84]. Therefore, the expression and activity of NET are key factors affecting the level of norepinephrine molecules binding with cardiac adrenergic receptors and maintenance of the intrinsic myocardial electrophysiology and contractility. 

In addition to cardiac postganglionic sympathetic neurons, satellite glial cells are also located in stellate ganglia. Satellite glial cells exist ubiquitously in peripheral ganglia, including sympathetic, parasympathetic, and sensory ganglia, which almost envelop peripheral ganglionic neuronal cell somata [85]. Although astrocytes, a counterpart of satellite glial cells in the central nervous system, have been widely studied, a few studies reported the morphology and function of satellite glial cells in peripheral ganglia, including stellate ganglia. Satellite glial cells are derived from the neural crest and have a relatively small volume with a thinner sheath and flattened processes [86]. Normally, satellite glial cells around a given neural cell body are in close contact with each other, which forms a neuron–glial unit to almost separate the connection between neuronal cells [85,86,87]. The distance between satellite glial cells and the membrane of peripheral ganglionic neuronal cells is about 20 nm, which is similar to that of the synaptic cleft [85]. This close synapse-like structural pattern could provide a structural basis for the neuron–satellite glial cell interaction, although little is known about the function of satellite glial cells in peripheral ganglia, especially stellate ganglia. 

Although satellite glial cells are electrically non-excitable without voltage-gated sodium and calcium channels, the inwardly rectifying potassium channels (Kir4.1) are expressed in satellite glial cells [88]. Satellite glial cells also express gap junction channels (such as connexin 43, Cx43) and purinergic 2 (P2) receptors (such as P2X and P2Y receptors) on the cell membrane [85,86]. Modulation of Kir4.1 permeability and activation of Cx43 and P2 receptors might depolarize the membrane of satellite glial cells and increase the intracellular calcium concentration to induce the release of excitatory mediators (such as ATP and some cytokines) from these cells for further activation of adjacent neurons [85,87,89]. Additionally, the release of nerve growth factor (NGF) from satellite glial cells may be involved in the maintenance and restoration of adjacent neurons [90]. Moreover, using single-cell RNA sequencing, one recent study demonstrated that the mature satellite glial cells in stellate ganglia are classed into five subpopulations of satellite glial cells with different functions of the subclusters [91]. Therefore, satellite glial cells in stellate ganglia could play an important role in the regulation of sympathetic neuronal function and maintenance of these adjacent neurons. 

## 3. Remodeling of Cardiac Postganglionic Sympathetic Neurons and Its Role in Cardiac Sympathetic Overactivation, Malignant Arrhythmias, and Cardiac Sudden Death in HF 

Although cardiac sympathetic remodeling can contribute to cardiac sympathetic overactivation and has not been systematically explored during HF progression, scattered information about HF-triggered cardiac sympathetic remodeling is demonstrated by most previous studies, including our work. These include structural and functional changes in cardiac postganglionic sympathetic cell somata and their nerve terminals.

### 3.1. Structural Remodeling in Cardiac Postganglionic Sympathetic Neurons Located in Stellate Ganglia

The majority of sympathetic nerves projecting to the heart originate in cardiac sympathetic postganglionic neurons located in stellate ganglia. There are limited data on the actual remodeling of cardiac sympathetic neuronal structures in stellate ganglia. One previous study demonstrated that stellate ganglionic nerve sprouting and density are elevated at one to four weeks after coronary artery ligation-induced rabbit myocardial infarction, which is mediated by nerve growth factor [92]. In a porcine chronic myocardial infarction model (six weeks after left anterior coronary descending artery occlusion-induced myocardial infarction), chronic myocardial infarction significantly increased the size of neuronal somata in the left stellate ganglion [93]. Using growth-associated protein 43, synaptophysin, and tyrosine hydroxylase as immunohistochemistry markers of synapses and sympathetic neurons in stellate ganglia, Han et al. found that the synaptic density and size of sympathetic neurons in the left stellate ganglion increased in dogs two months post-myocardial infarction [94]. Tan et al. also reported that the sympathetic nerve density (immunoreactivity of tyrosine hydroxylase) in stellate ganglia was markedly increased in canines 12 weeks after premature ventricular contraction-induced cardiomyopathy [95]. Similarly, Ajijola et al. demonstrated that the size of stellate ganglionic neurons increased with an increase in the neuronal adrenergic phenotype and neuropeptide Y-positive neurons in pigs at 6 weeks post left circumflex or right coronary artery occlusion-induced myocardial infarction [96]. In humans with cardiomyopathy, the size of the stellate ganglionic neurons is significantly increased without ganglionic fibrosis and changes in the neuronal density (cell number/tissue area) and synaptic density [97]. Although these studies in myocardial infarction-induced animal HF models and humans with cardiomyopathy are not totally consistent and it is unclear how structural alterations of cardiac postganglionic sympathetic neurons affect the progression and prognosis of HF, the morphological changes in these neurons could be associated with increased stellate ganglionic nerve activities and further related to cardiac sympathetic overactivation and malignant arrhythmias. Certainly, investigation of the structures of subcellular organelles (including nucleus, mitochondria, lysosome, and secretory vesicles) by electron microscopy is necessary to explore the cellular and molecular mechanisms underlying the structural remodeling of cardiac postganglionic sympathetic neuronal somata in HF.

### 3.2. Functional Remodeling in Cardiac Postganglionic Sympathetic Neurons Located in Stellate Ganglia

The function of neurons is to transmit electrical signals over long distances through the generation of action potentials. The left stellate ganglionic nerve activity is increased in ambulatory dogs with pacing- or coronary artery occlusion-induced HF [94,98]. Tu et al. demonstrated that the cell excitability in cardiac postganglionic sympathetic neurons located in stellate ganglia increases in coronary artery ligation-induced HF rats [99]. Although various types of ion channels (such as voltage-gated sodium, calcium, and potassium channels) can contribute to the generation of action potentials in cardiac postganglionic sympathetic neurons, voltage-gated calcium channels should be considered as the mechanism governing the increased cell excitability of these sympathetic neurons in HF because calcium influx through voltage-gated calcium channels is a key trigger for the release of neurotransmitters from neuronal nerve terminals [100,101,102]. There are five subtypes of voltage-gated calcium channels (T, L, N, P/Q, and R) functionally characterized in central and peripheral neurons [103,104]. A pore-forming α-subunit in all subtypes of calcium channels determines the biophysical and pharmacological properties of calcium channels [105]. There are three major families of α-subunits: (1) Cav1 (Cav1.1, Cav1.2, and Cav1.3) encodes L-type calcium channels; (2) Cav2 encodes P/Q (Cav2.1), N (Cav2.2), and R (Cav2.3) types of calcium channels; and (3) Cav3 encodes T-type calcium channels [105,106]. In fact, Tu et al. reported that the L, N, P/Q, and R types of calcium channels are expressed in cardiac postganglionic sympathetic neurons [99]. However, HF only increases N-type calcium currents and does not affect the mRNA and protein expression of all calcium channel subtypes in these sympathetic neurons [99]. Some previous studies demonstrated that N-type calcium channels, predominantly expressed in the nervous system, play an important role in modulating neurotransmitter release at sympathetic neve terminals [107,108]. More importantly, increased N-type calcium currents in cardiac postganglionic sympathetic neurons contribute to the elevated cell excitability of these neurons, cardiac sympathetic overactivation, and malignant arrhythmias in HF [109,110]. Until now, there has been no report about the involvement of other ion channels in HF-increased cell excitability of cardiac postganglionic sympathetic neurons. 

### 3.3. Structural Remodeling in Cardiac Postganglionic Sympathetic Nerve Terminals

Cardiac sympathetic nerve terminals are directly embedded in the myocardium with a heterogeneous distribution. Most previous studies reported the information about structural remodeling in cardiac postganglionic sympathetic nerve terminals during HF progression from acute myocardial infarction to chronic HF, with no consistent conclusion. Acute myocardial infarction could result in sympathetic nerve terminal denervation in the scar and viable myocardium beyond the infarcted area [111,112,113]. Then, the regeneration of sympathetic nerve terminals in the heart has been characterized by nerve spouting and a high density of nerve fibers in the periphery of the necrotic myocardium of failed hearts [114,115,116]. Additionally, some previous studies also reported cardiac sympathetic nerve terminal denervation in HF [117,118]. Regions of cardiac sympathetic nerve terminal denervation and hyperinnervation are present in the same failed heart to form the heterogeneity of the cardiac sympathetic nerve distribution [119,120]. Iodine-123 meta-iodobenzylguanidine (^123^I-MIBG) or other radiolabeled neurotransmitter analogs (including the recently used F-18 meta-fluorobenzylguanidine) and cardiac neurotransmission imaging with single-photon emission computed tomography (SPECT) and positron emission tomography (PET) have been employed to noninvasively assess the integrity of human NET and further evaluate cardiac sympathetic nerve innervation [121,122,123]. However, poor imaging quality, difficulty in distinguishing different cardiac structures, and high cost limit this technique’s application in animal studies, especially small animal studies. Additionally, previous studies used immunohistochemical staining in several myocardial slices to evaluate the structural remodeling of cardiac sympathetic nerve terminals, which cannot represent the distribution of cardiac sympathetic nerve terminals in the whole heart with a neglected heterogeneous distribution of nerve terminals. Although these structural alterations of cardiac postganglionic sympathetic nerve terminals are considered to create a high-yield substrate for malignant arrhythmias in HF [124], the conclusion from these previous studies should be questioned. Using three-dimensional assessment of the cardiac sympathetic network in cleared transparent murine hearts, one recent study demonstrated both cardiac sympathetic nerve terminal hyperinnervation and denervation in the whole heart at 2 weeks post myocardial infarction [125]. It is not clear whether the same phenomenon (sympathetic nerve terminal hyperinnervation and denervation) is also present in the whole heart with HF using three-dimensional assessment of the cardiac sympathetic network. Therefore, the timing and patterns of the cardiac sympathetic nerve terminal remodeling in HF should be re-evaluated in future studies. Indeed, structural remodeling and norepinephrine release in cardiac postganglionic sympathetic nerve terminals in HF should be combined to assess the association of cardiac sympathetic activation and malignant arrhythmias because it is unclear whether reinnervated cardiac sympathetic nerve terminals can release norepinephrine like mature sympathetic nerve terminals in the heart.

### 3.4. Functional Remodeling in Cardiac Postganglionic Sympathetic Nerve Terminals

The function of cardiac sympathetic nerve terminals is to release neurotransmitters, including norepinephrine, which bind to adrenergic receptors to regulate cardiac function in physiological and pathophysiological conditions. Cardiac norepinephrine spillover is measured by calculating the amount of plasma norepinephrine entering the heart and the amount of norepinephrine exiting the heart. An elevation in cardiac norepinephrine spillover occurs in HF [126,127,128,129,130,131], which primarily results from the increase in cardiac norepinephrine synthesis and release, and the decrease in norepinephrine reuptake [84,132]. By ^123^I-MIBG with SPECT and PET images, some clinical studies measured the heart-to-mediastinum (H/M) ratio and ^123^I-MIBG washout (WO) rate to demonstrate an increased level of sympathetic neurotransmitter in HF [133,134,135,136,137]. Although the above studies indirectly tested norepinephrine release and demonstrated that these measured scientific parameters are strong predictors of cardiac sympathetic overactivation, heart failure progression, life-threatening arrhythmias, and cardiac sudden death in HF, there is limited information available on the direct measurement of norepinephrine release from cardiac sympathetic nerve terminals during HF progression. In vivo cardiac microdialysis with HPLC can directly test norepinephrine release from cardiac sympathetic terminals [138,139,140], but it is hard to obtain stable data of norepinephrine release due to the heterogeneous distribution of cardiac sympathetic nerve terminals as described above. Zhang et al. recently reported an electrochemistry recording, patch-clamp technique with a carbon fiber electrode for the catecholamine release from adrenal chromaffin cells [141]. The development of this recording in in vivo cardiac slices could be an innovative technique for the direct measurement of norepinephrine release from cardiac sympathetic nerve terminals, which possibly avoids the interference of the heterogeneous distribution of cardiac sympathetic nerve terminals and also analyzes norepinephrine release kinetics (including the maximal amplitudes of norepinephrine release and reuptake).

Cardiac sympathetic activation is dependent on two major components, namely circulating catecholamines from the adrenal medulla and local norepinephrine release from cardiac postganglionic sympathetic nerve terminals. Stellate ganglion stimulation, including left, right, or bilateral stellate ganglion stimulation, produces distinct patterns of cardiac myocyte repolarization in the normal porcine heart, evaluated by the analysis of epicardial and endocardial electrograms, whereas marked dispersion of cardiac myocyte repolarization does not occur when exogenous norepinephrine is infused (circulating norepinephrine) [142]. From these data, it has been demonstrated that stellate-ganglion-stimulated dispersion of cardiac myocyte repolarization is highly arrhythmogenic, compared to the more uniform changes in cardiac myocyte repolarization triggered by circulating norepinephrine [142,143]. As described above, cardiac sympathetic denervation, a key approach to the current therapy of HF, highlights the role of cardiac sympathetic remodeling in cardiac sympathetic overactivation, malignant arrhythmias, and cardiac sudden death in HF [38,42,43]. Using in vivo shRNA transfection into stellate ganglia, Zhang et al. demonstrated that ion channel remodeling in cardiac postganglionic sympathetic neurons is involved in cardiac sympathetic overactivation and ventricular arrhythmogenesis in coronary artery ligation-induced HF [110]. 

Recent studies have reported that elevated neuropeptide Y and other sympathetic co-transmitters released from cardiac sympathetic neurons act on neuropeptide Y or other related receptors on the membrane of cardiac myocytes to cause the development of HF and ventricular arrhythmias [144,145,146,147,148]. As a result, cardiac postganglionic sympathetic remodeling, including alterations of the sympathetic co-neurotransmitter release (such as norepinephrine, neuropeptide Y, and galanin), could be associated with cardiac sympathetic overactivation, malignant arrhythmias, and cardiac sudden death in HF.

## 4. Mechanisms Underlying the Remodeling of Cardiac Postganglionic Sympathetic Neurons in HF

The mechanisms responsible for the remodeling of cardiac postganglionic sympathetic neurons in HF are not well understood and could be multifactorial. Additionally, cardiac postganglionic sympathetic nerve terminals are embedded in the myocardium. Therefore, the microenvironment surrounding cardiac postganglionic sympathetic nerve terminals in the myocardium should be the key factor for the structural and functional remodeling of these nerve terminals, although the factors that modulate cardiac postganglionic sympathetic cell somata might also affect their nerve terminals. 

### 4.1. Mechanisms Associated with the Remodeling of Cardiac Postganglionic Sympathetic Cell Somata

Nerve growth factor (NGF), a prototypical member of the neurotrophin family, is normally involved in the maintenance, proliferation, and survival of neurons. The action of NGF in targeted cells is initiated by its high-affinity binding to the tropomyosin receptor kinase A (TrkA, also named neurotrophic tyrosine kinase receptor 1) receptors in mature sympathetic neurons [149,150]. NGF is usually produced by sympathetic innervated organ/tissues. A high level of NGF is present in stellate ganglia, in which NGF is accumulated by retrograde axonal transport to affect the function of neurons [151,152]. The local production of NGF by satellite glial cells in stellate ganglia could be another source because NGF mRNA is expressed in neurons and satellite glial cells from trigeminal ganglia [90]. Although a high level of NGF is probably involved in the structural and functional remodeling of cardiac postganglionic sympathetic cell somata, including an increased cell size of sympathetic neurons, synaptic density, and neuronal excitability in stellate ganglia in HF, further studies are needed to provide direct evidence. 

As mentioned above, satellite glial cells exist ubiquitously in peripheral ganglia, including sympathetic, parasympathetic, and sensory ganglia. In addition to the local production of NGF by satellite glial cells in stellate ganglia, satellite glial cell–macrophage communication could be another mechanism responsible for the remodeling of cardiac postganglionic sympathetic neurons in HF. One recent study demonstrated that acute or chronic intestinal inflammation activates enteric glial cells to release macrophage colony-stimulating factor (M-CSF) through the connexin-43 hemichannel-cytosolic PKC-MAPK-cell membrane TNFα-converting enzyme (TACE) signaling pathway [89]. M-CSF is a key factor for the regulation of macrophage survival, proliferation, migration, and activation through binding with M-CSF receptors on the macrophage membrane [153,154]. Chronic inflammation, with activation of both cytokines and immune cells (such as macrophages), is a major feature of HF [155,156,157]. Macrophages play a key role in mediating inflammatory responses in the post-myocardial infarction heart [158,159]. Our recent study already found that elevation of cytokines and macrophages in stellate ganglia is involved in cardiac sympathetic overactivation and ventricular arrhythmogenesis in HF [160]. Growing evidence suggests that inflammation-raised cytokines increase the expression and activation of cyclin-dependent kinase 5 (CDK5, a proline-directed serine/threonine kinase) in some tissues and cell lines [161,162,163]. CDK5 can phosphorylate the N-type calcium channels and the latter induce cardiac sympathetic overactivation and ventricular arrhythmias in HF [110,164,165]. Therefore, further understanding of the relations among satellite glial cells, macrophages, and cardiac postganglionic sympathetic neurons in stellate ganglia can provide therapeutic targets against cardiac sympathetic overactivation and malignant ventricular arrhythmias in HF. 

Oxidative stress is also considered to be another factor for the remodeling in cardiac sympathetic neurons in HF. Ajijola et al. found that stellate ganglia from patients with cardiomyopathy and arrhythmias exhibit oxidative stress [166]. In small animal models of myocardial infarction-induced HF, oxidative signaling is increased in stellate ganglia [167]. Therefore, the effect of oxidative stress on the remodeling in cardiac sympathetic neurons and interaction between oxidative stress and inflammation-raised cytokines in HF should be explored in future studies.

### 4.2. Mechanisms Associated with the Remodeling in Cardiac Postganglionic Sympathetic Never Terminals

As described above, NGF is mainly produced in the sympathetic innervated organs/tissues. Western bolt analysis in the left ventricle demonstrated an elevation of NGF in explanted, failing human hearts compared to normal, donor hearts [168]. NGF overexpression in the sympathetic targeted organs/tissues causes sympathetic nerve hyperinnervation, which could be involved in nerve spouting and a high density of sympathetic nerve terminals in the periphery of the necrotic myocardium of failed hearts [116,149,152,169]. When NGF binds with TrkA receptors on sympathetic nerve terminals, activated TrkA receptors regulate cytoskeletal dynamics through successive activation of the MAPK and PI3K-Akt pathways, and endocytosed TrkA receptors promote sympathetic nerve growth and hyperinnervation through a calcineurin-dynamin 1 signaling pathway [170,171,172]. Although NGF could be considered as the key factor modulating sympathetic nerve innervation in targeted organs/tissues in physiological and pathophysiological conditions, some other factors also contribute to cardiac sympathetic hyperinnervation in either an independent style or by association with NGF [149,152]. These endogenous factors include growth differentiation factor 5 (GDF5), TNF receptor 1, leukemia inhibitory factor, cardiotrophin-1, and leptin [173,174,175,176]. Additionally, pro-NGF, pro-brain-derived neurotrophic factor (pro-BDNF), and brain-derived neurotrophic factor (BDNF) activate the p75 neurotrophin receptors (p75NTRs) and death receptor 6 (DR6), two members of the TNF super-family, to stimulate sympathetic nerve denervation [177,178,179,180,181,182,183], which occurs in the cardiac infarcted area and myocardium adjacent to the scar in myocardial infarction-induced HF. 

Using a mouse model of cardiac ischemia-reperfusion, one recent study demonstrates that therapeutics-restored sympathetic reinnervation of the infarcted area decreases M1-like macrophages and elevates the numbers of dendritic cells, M2-like macrophages, and Treg cells [184]. There are different contributions of M1-like and M2-like macrophages to cardiac sympathetic remodeling. Therefore, future studies are needed to assess the interaction between cardiac sympathetic remodeling and the different types of macrophages in HF 

Functional remodeling in cardiac postganglionic sympathetic nerve terminals includes the change in norepinephrine synthesis, release, and reuptake in HF. For norepinephrine synthesis, normally, tyrosine is converted to 3,4-dihydrooxyphenyl alanine (DOPA) by the rate-limiting enzyme, tyrosine hydroxylase (TH). DOPA is then converted to dopamine by L-aromatic acid decarboxylase. After that, the vesicular monoamine transporter translocates dopamine into storage vesicles, in which dopamine is converted to norepinephrine by dopamine β-hydroxylase. Although it has been shown that cardiac norepinephrine synthesis is increased in HF [132], so far, it is unclear which enzyme and potential mechanism(s) are responsible for the increase in cardiac norepinephrine synthesis in HF. For norepinephrine release from sympathetic nerve terminals, Huang et al. demonstrated that the action potential modulates calcium-dependent and -independent (voltage-dependent) quantal norepinephrine release in the mammalian sympathetic nervous system [185]. Our recent studies found that the activated macrophage-triggered increase in the N-type calcium currents results in neuronal overexcitation of cardiac postganglionic sympathetic neurons in a myocardial infarction-induced HF animal model [110,160], which could contribute to the increase in the norepinephrine release from cardiac sympathetic nerve terminals in the HF state. For norepinephrine reuptake, it is responsible for the rapid removal of interstitial norepinephrine through the norepinephrine transporter (NET) after norepinephrine release from cardiac sympathetic nerve terminals. Much evidence confirms a reduction in the neuronal NET density and activity at cardiac sympathetic nerve terminals in the failing heart [186,187,188]. In cultured rat neuroblastoma cells (PC12 cell line), exogenous norepinephrine significantly reduces the expression of NET protein but not NET mRNA [189,190], which suggests post-transcriptional downregulation for NET in HF [82]. Normally, NET located in the cell membrane is regulated by glycosylation [191]. Therefore, the effect of norepinephrine on the reduction in the neuronal NET density in HF could be associated with endoplasmic reticulum stress-reduced glycosylation, causing the trafficking of NET to the cell membrane [190]. Additionally, endothelin binding with endothelin receptors also inhibits NET activity in HF by affecting NET phosphorylation because the NET activity is, in the short and long term, modulated by protein kinase A, C, and G and calcium-calmodulin-dependent protein kinase [191,192]. It is unclear whether NET activity and expression are modulated by other endogenous factors in HF, such as the renin-angiotensin-aldosterone system, bradykinin, nitric oxide, natriuretic peptides, etc. 

## 5. Conclusions

In this review, we updated the information about the cardiac sympathetic remodeling in stellate ganglia and potential mechanisms and the role of cardiac sympathetic remodeling in cardiac sympathetic overactivation, arrhythmias, and cardiac dysfunction in HF (Figure 2). Using droplet-based high-throughput single-cell RNA sequencing, one recent study separated eight large clusters in the superior cervical ganglion (one type of peripheral sympathetic ganglia) from young adult mice based on the expression of canonical marker genes [193]. These clusters include satellite glial cells (Plp1, Fabp7), Schwann cells (Plp1, Ncmap), sympathetic neurons (Snap25, Th), vascular endothelial cells (Ly6c1), macrophages (C1qb), T cells (Trb2), fibroblasts (Dcn), and mural cells (Rgs5) [193]. Following the development of innovative techniques, including epigenetics, transcriptomics, proteomics, optogenetics, single-cell RNA sequence analysis, etc., more details on stellate ganglionic cell remodeling with related mechanisms in HF will be explored. Stellate ganglia could be a therapeutic target against cardiac sympathetic overactivation and myocardial electrophysiological and contractile dysfunction in HF.

## Figures and Tables

**Figure 2 ijms-23-13311-f002:**
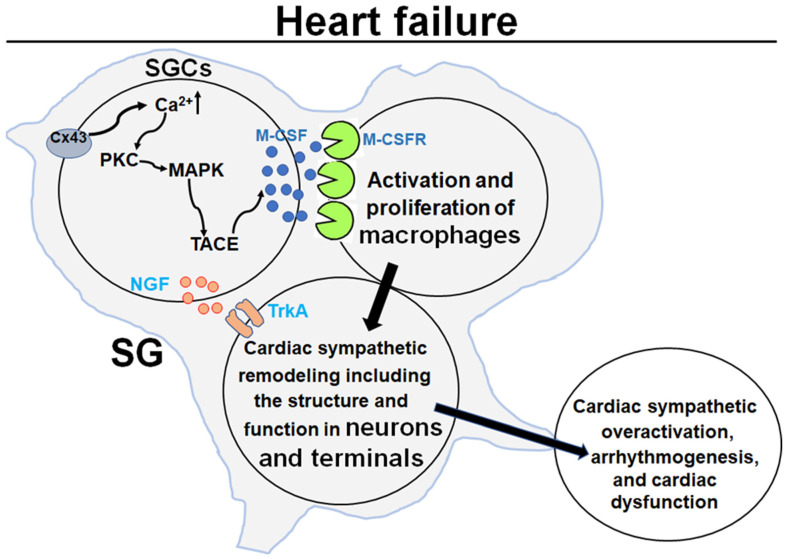
Mechanisms underlying the remodeling of cardiac postganglionic sympathetic neurons in HF. SGC: satellite glial cell; Cx43: connexin 43; PKC: protein kinase C; MAPK: mitogen-activated protein kinase; TACE: tumor necrosis-converting enzyme; M-CSF; macrophage colony-stimulating factor; M-CSFR: macrophage colony-stimulating factor receptor; NGF: nerve growth factor; TrkA: tropomyosin receptor kinase A; SG: stellate ganglion.

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
