# Peer review of "Stellate Ganglia and Cardiac Sympathetic Overactivation in Heart Failure"

_ijms, 2022, doi:10.3390/ijms232113311_

Round 1
Reviewer 1 Report
Normally sympathetic activation of the heart results in an increase in heart rate and inotropy through released norepinephrine binding with beta1-adrenergic receptors. However, prolonged sympathetic overactivation exacerbates heart failure and triggers the development of heart failure. Although we understand that three components of the cardiac sympathetic system, including afferent component, central commend, and efferent component, could be involved in cardiac sympathetic overactivation in heart failure, the detail mechanisms are unclear. This review manuscript clearly introduced anatomy and physiology of stellate ganglia, remodeling of cardiac postganglionic sympathetic neurons and its role in cardiac sympathetic overactivation in heart failure, and potential mechanisms responsible for the remodeling of cardiac sympathetic neurons in heart failure. This is an interesting review manuscript. The manuscript could give readers widespread and profound information in this research area, especially logical interpretation based on the previous research papers. I have some comments.
1. The author mentioned that elevation of cytokines and macrophages in stellate ganglia is involved in cardiac sympathetic overactivation and ventricular arrhythmogenesis in heart failure. Could the author provide more detail information how (what signaling pathways) elevation of cytokines and activation of macrophages cause cardiac sympathetic overactivation in heart failure?
2. As toxic by-products of aerobic metabolism, reactive oxygen species (ROS) plays an important signaling role in a large number of pathophysiological conditions. Please give the information if ROS is also involved in the remodeling of cardiac postganglionic sympathetic neurons.
Author Response
We are grateful to the reviewer for positive and thoughtful comments. We have revised our manuscript in response to these suggestions. All revisions are highlighted in red in the revised manuscript. We have also point-to-point responded the reviewer's comments.
1. The author mentioned that elevation of cytokines and macrophages in stellate ganglia is involved in cardiac sympathetic overactivation and ventricular arrhythmogenesis in heart failure. Could the author provide more detail information how (what signaling pathways) elevation of cytokines and activation of macrophages cause cardiac sympathetic overactivation in heart failure?
Response: Thanks for the reviewer’s suggestion. We addressed this issue in the revised manuscript (last 5 lines in page 14).
2. As toxic by-products of aerobic metabolism, reactive oxygen species (ROS) plays an important signaling role in a large number of pathophysiological conditions. Please give the information if ROS is also involved in the remodeling of cardiac postganglionic sympathetic neurons.
Response: We added the information about relationship between oxidative stress and the remodeling of cardiac sympathetic neurons in the revised manuscript (the second paragraph in page 15).
3. Response: although the reviewer required extensive editing of English and style in Review Report Form, we checked our manuscript again and did not think it is necessary. We only corrected some words in the revised manuscript.
Reviewer 2 Report
The authors present a well-written review article that consolidates a wide-range of studies providing evidence for sympathetic hyperactivity, and altered signalling via the stellate ganglia in heart failure. The addition and expansion of a couple of areas would strengthen the review:
1. the paper mentions sympathetic co-transmitters but does not elaborate on their role in the pathophysiology of heart failure. Including studies on NPY and its role in coronary reperfusion post-MI and some discussion of other neurotransmitters would be beneficial.
2. The discussion on sympathetic reinnervation should include recent work by the Habecker group (JACC:BTS Sept. 2022), making the section up-to-date and topical.
Author Response
We are grateful to the reviewer for positive and thoughtful comments. We have revised our manuscript in response to these suggestions. All revisions are highlighted in red in the revised manuscript. We have also point-to-point responded the reviewer's comments.
1. the paper mentions sympathetic co-transmitters but does not elaborate on their role in the pathophysiology of heart failure. Including studies on NPY and its role in coronary reperfusion post-MI and some discussion of other neurotransmitters would be beneficial.
Response: Thanks for the reviewer’s suggestion. We added the role of sympathetic co-neurotransmitters in the pathophysiology of heart failure in the revised manuscript (the second paragraph in page 13)
2. The discussion on sympathetic reinnervation should include recent work by the Habecker group (JACC:BTS Sept. 2022), making the section up-to-date and topical.
Response: We added the work from Dr. Habecker’s group (Sepe, et al. JACC:BTS 2022, 7, 915-930) in the revised manuscript (the second paragraph in page 16).